# Sentinel Lymph Node Biopsy in Uterine Cancer: Time for a Modern Approach

**DOI:** 10.3390/cancers15020389

**Published:** 2023-01-06

**Authors:** Enora Laas, Virginie Fourchotte, Thomas Gaillard, Léa Pauly, Fabien Reyal, Jean-Guillaume Feron, Fabrice Lécuru

**Affiliations:** 1Service de Chirurgie Sénologique, Gynécologique et Reconstructrice, Institut Curie, 26 Rue d’Ulm, 75005 Paris, France; 2Faculté de Médecine, Université de Paris Cité, 75006 Paris, France; 3Residual Tumor & Response to Treatment Laboratory, RT2Lab, INSERM, U932 Immunity and Cancer, Institut Curie, Université Paris, 75005 Paris, France

**Keywords:** sentinel lymph node biopsy, cervical cancer, endometrial cancer

## Abstract

**Simple Summary:**

The sentinel lymph node biopsy technique is widely used in uterine cancers. Some new parameters as ultrastadification or molecular biology have modified our algorithms which are becoming more and more complex. We discuss here the integration of the sentinel node biopsy technique in the global strategy of management of uterine cancers.

**Abstract:**

Since the validation of the sentinel node technique (SLN) for vulvar cancer 20 years ago, this technique has been introduced in the management of operable cervical cancer and endometrial cancer. For cervical cancer a “one fits all” attitude has mainly been presented. However, this approach, consisting of a frozen section during the operation, can be discussed in some stages. We present and discuss the main option for each stage, as well as some secondary possibilities. For endometrial cancer, SLN is now the technique of choice for the nodal staging of low- and intermediate-risk groups. Some discussion exists for the high-risk group. We also discuss the impacts of using preoperatively the molecular classification of endometrial cancer. Patients with POLE or TP53 mutations could have different nodal staging. The story of SLN in uterine cancers is not finished. We propose a comprehensive algorithm of SLN in early cervical and endometrial cancers. However, several ongoing trials will give us important data in the coming years. They could substantially change these propositions.

## 1. Introduction

The sentinel lymph node (SLN) biopsy (SLNB) technique was introduced 20 years ago in gynaecologic oncology, aiming to limit the morbidity of nodal staging in vulvar cancer. It has since become increasingly popular among the gynaecologic oncology community, especially in endometrial cancer and, more recently, in cervical cancer.

This approach provides several improvements to daily practice. First, the surgical techniques have been established to ensure a reliable sensitivity and negative predictive value. Second, the prognostic importance of isolated tumour cells (ITC) and micrometastases have been addressed, conferring to this technique a prognostic input beyond only a reduction in morbidity. Third, new markers, such as IndoCyanine Green, have been validated.

The integration of SLNB into uterine cancer decisional trees has been less addressed. Several changes occurred in the knowledge and management of gynaecological cancers during the past years. Cervical cancer surgery has deeply evolved, with the implementation of fertility sparing surgery, less radical surgery, and more personalized operations. For endometrial cancer, the molecular classification has revolutionized the risk evaluation and treatment decision making. It appears now necessary to clearly establish the position of SLNB in the different algorithms. We reviewed the standards and pending questions of SLNB in cervical and endometrial cancer, according to published data and ongoing trials.

## 2. Cervical Cancer (FIGO 2018 Classification)

The actual place of SLNB in the management of operable cervical cancer must be defined according to different and sometimes conflicting aims and limitations:SLNB performance in cervical cancer. The diagnostic accuracy of SLNB has been demonstrated for squamous and adenocarcinomas (+adenosquamous) measuring less than 4 cm in largest diameter [1,2]. The negative predictive value has been considered as a whole for all stages together. However, diagnostic accuracy differs between tumours < 2 cm vs. 2–4 cm lesions [3].To ensure high and bilateral detection rates, a full dissection of paravesical and pararectal fossa are necessary on both sides [4]. The aim is to pick up only the first node linked to the cervix by a lymphatic channel on both sides. This ensures that the “true” SLN is sampled and not a second echelon node. This step fruitfully prepares the radical hysterectomy or parametrial dissection in the case of a “one step” radical operation after negative frozen section (FS) of SLN. Conversely, a “two steps” surgery, requiring a parametrium dissection some days or weeks after the SLN sampling, exposes patients to a risk of operative difficulties and perioperative complications due to tissue inflammation and fibrosis.Postoperative radiation therapy after radical hysterectomy is highly toxic and should be avoided. [5]. However, this issue should be revaluated with recent data, taking into account modern radiotherapy [6,7].Resection of parametrium is no more necessary for all cases.Isolated tumour cells and micrometastases are accurately diagnosed by definitive pathological examination with serial sectioning and immune-histo-chemistry [8]. FS misses most ITC, several micrometastases, and some small macrometastases. The sensitivity is 50–60% for diagnosis of macro and micrometastases in most series [9,10].Prognostic of low-volume metastases is still under evaluation. However, most data and authors today consider that micrometastases share the same prognostic effect than macrometastases [11,12,13]. Role of ITC is not certain today.The SLN technique also allows for exploration of the parametrium, searching for paracervical nodal spread [14]. Moreover, in tumours < 20 mm and with negative SLN after ultrastaging, parametrial involvement occurs in <1% of cases, and less radical surgery may be a realistic option for these patients [15].Rate of nodal positivity increases with stage and presence of lympho vascular space invasion (LVSI). Nodal metastases are quite rare for stage Ia1 with lymphovascular emboli (13%), and more common for Ib tumours (12%) (this integrates macrometastases and micrometastases or isolated tumour cells (ITC)) [16,17]. The place of SLN in the algorithm will depend on the prioritization of negative predictive value (utmost for fertility sparing surgery) and the necessity of parametrial resection.

Considering these parameters, we have tried to define the best place of SLNB in the operable cervical cancer decisional tree.

### 2.1. Stage Ia1 with Lymphovascular Emboli (LVSI) and Stage Ia2 (Figure 1)

These stages account for 8% of early cervical cancer. They are diagnosed on a cone biopsy. A complete excision of the lesion is recommended and parametrial resection is not indicated in ESGO guidelines (conisation “in sano” or hysterectomy) [18]. NCCN guidelines accept “cone biopsy with negative margins” as well as “radical trachelectomy” [19]. In the ConCerv trial, the cumulative incidence of recurrence was 3.5% at 2 years, showing that radical surgery is of limited interest in these patients [20].

**Figure 1 cancers-15-00389-f001:**
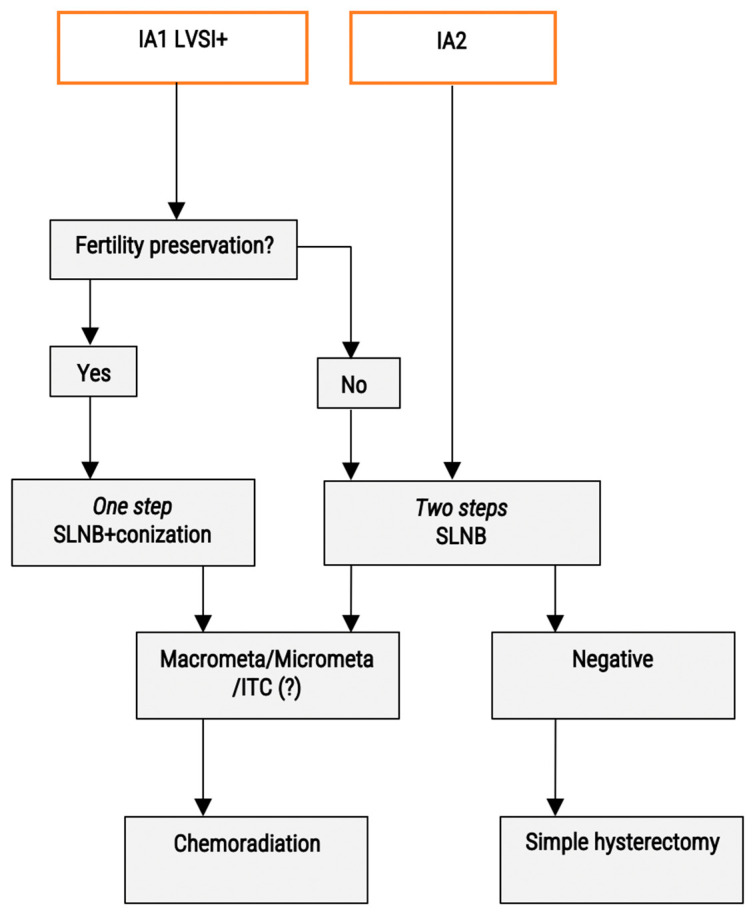
Proposed algorithm for the nodal management of early stage cervical cancer—Stages IA. LVSI: lympho-vascular space invasion; ITC: isolated tumour cells; SLND: sentinel lymph node biopsy.

Concerning the lymph node staging, it “can be considered” for the stage Ia1 disease; “should be considered” for the stage Ia2 disease; and “SLN biopsy without additional pelvic lymph node dissection is an acceptable method” [19]. NCCN guidelines still recommend pelvic lymphadenectomy and consider SLN an option. In the SENTICOL I and II cohorts, the final rate of nodal positivity was 17%, including ITC, micro- and macrometastases [16,17]. In the ConCerv trial, which included Ia2 and Ib1 patients, the rate of nodal positivity was 5% but low volume disease was not clearly assessed.

In patients wishing to preserve their fertility, the knowledge of nodal spread is of utmost importance, and the low sensitivity of FS for diagnosis of ITC and especially micrometastases is not acceptable. However, this does not preclude the conisation, and a one-step procedure is logical (conization + SLN biopsy). Precise examination of the parametrium +/− dissection for searching parametrial SLN could be carried out in these cases [14]. Rare cases with macro- or micrometastases will be referred secondarily to chemoradiation, without impairing the functional prognosis. Concerning patients with ITC, no strong survival data exists. Case discussion should be carried out in this situation.

In older patients, or women not wishing for a fertility sparing approach, a non-radical hysterectomy can be proposed after a full pathological analysis of the SLNs. A two-steps procedure appears appropriate. Patients without nodal spread will undergo a simple hysterectomy. Node-positive patients will be referred to chemoradiation. This schema will avoid post-hysterectomy radiation therapy.

### 2.2. Stage Ib1 (Figure 2)

These patients constitute the most important part of operable cervical cancers (81%).

**Figure 2 cancers-15-00389-f002:**
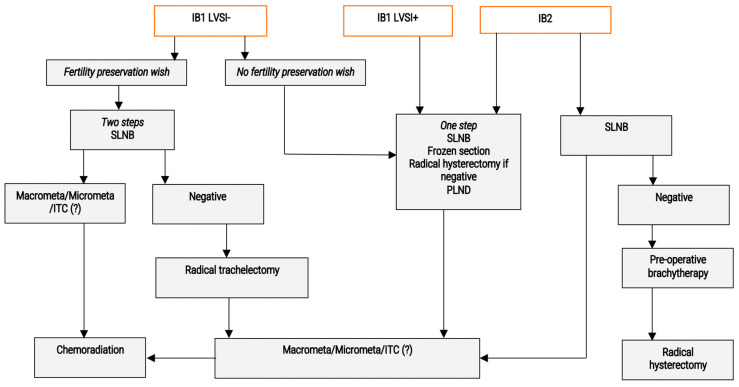
Proposed algorithm for the nodal management of early stage cervical cancer—Stages IB. LVSI: lympho-vascular space invasion; ITC: isolated tumour cells; SLNB: sentinel lymph node biopsy; PLND: para-aortic lymph node dissection.

#### 2.2.1. Stage Ib1 without LVSI

To date, radical hysterectomy or radical trachelectomy remains the standard of care for these patients, when the absence of nodal spread is demonstrated. A systematic bilateral parametrial dissection and resection with safe margins is then necessary. The second objective of the treatment is to avoid the association of radical surgery and postoperative radiation therapy. Concerning the nodal staging, a standard lymphadenectomy is recommended, performed at the first step of surgical management with FS in ESGO guidelines. SLN biopsy is also recommended but should be completed by classical dissection. NCCN recommend a pelvic +/− paraaortic lymphadenectomy and consider SLN biopsy as an option only.

In patients with no fertility wish, a one-step procedure appears logical, with SLN biopsy, FS, and radical hysterectomy in the case of negative FS. The place of systematic complementary pelvic dissection is unclear, since diagnostic accuracy is good in this population. This strategy will make postoperative radiation therapy be indicated in 10% of patients with false negative FS [10]. In patients with a fertility wish, the alternative could be a two-step approach, with SLN biopsy and subsequent radical trachelectomy some days later, but only in patients without macro- or micrometastases. However, this approach will expose all patients to two general anaesthesia, operative difficulties, and complications due to iterative parametrial dissections. 

In the future, the results of the SHAPE trial could deeply change this algorithm. If parametrial resection is demonstrated to be non-necessary in this subset of patients, a two-step procedure will be more logical to limit the risk of postoperative radiation therapy.

#### 2.2.2. Stage Ib1 with LVSI

These patients have a higher risk of nodal metastases. ITC, micrometastases, and macrometastases are diagnosed in 3.3, 4.3, and 3.7% of cases, respectively [2,17]. Fertility-sparing surgery is contra-indicated in several teams. Here, again, a parametrectomy is necessary for the treatment of these lesions. A one-step procedure appears preferable, despite the necessity of postoperative radiation therapy in a subset of patients having micro- or macrometastases that were missed by FS.

Fertility-sparing treatment is not discussed for this stage, as it is not standard of care.

### 2.3. Stage Ib2, IIa1, IIa2 (Figure 2)

These stages are less frequent (10%). They are at the highest risk of nodal metastases. The reasoning of the stage IB1 LVSI+ can be used. Parametrial resection is necessary, and this militates against a one-step procedure: SLN biopsy, FS, and radical hysterectomy in case of negative FS. This will necessitate 17% of postoperative radiation therapy in operated patients with known morbidity. Conversely, chemoradiation will be offered in patients with nodal extension diagnosed at FS.

In some countries, preoperative brachytherapy is proposed for patients with Ib1 LVSI+, Ib2, and IIa1. This approach provides a complete macroscopic tumour response in 80% of cases. Two options can be proposed for these patients. The first one is performing primary brachytherapy, followed by SLN biopsy + FS +/− radical hysterectomy 6 to 8 weeks later. The second one is a serial steps management, with SLN biopsy first, brachytherapy in node negative patients, and finally non-radical hysterectomy. Patients with insufficient response or negative prognostic factors on the hysterectomy specimen will require radiation therapy.

### 2.4. Stage IIb and More

These patients are not a classical indication of SLNB. Laparoscopic paraaortic lymphadenectomy has been used for their nodal staging. Despite a recent trial showing no survival benefit of surgical staging when compared to imaging, several teams still perform this operation (UTERUS 11). An adapted technique, with harvest of second echelon SLN, could be an option in these patients to obtain the nodal status without a full dissection.

These propositions clearly demonstrate that we need improvements in several aspects of the technique. FS is poorly accurate and limits the efficacy of the technique in patients at the highest risk of nodal metastases. Diagnosis of metastatic nodes by imaging could provide the same information at lower costs.

## 3. Endometrial Cancer (Figure 3)

Lymph node status is crucial in early-stage endometrial cancer (EC), as it is part of the decision in adjuvant treatment. As systematic lymphadenectomy did not show significant improvements in survival compared to SLN dissection [21,22], and is associated with substantially lower risk of post-operative morbidity, SLN mapping is currently recommended for apparent low and intermediate risk [23]. In these groups, the risk of lymph node (LN) involvement is 2–6%, and half of them would be identified by pathologic ultrastaging, supporting the use of this technique [24]. Conversely, patients with tumours confined to the endometrium (without myometrial invasion) did not have any positive sentinel lymph nodes and lymph node exploration could be omitted in these cases.

**Figure 3 cancers-15-00389-f003:**
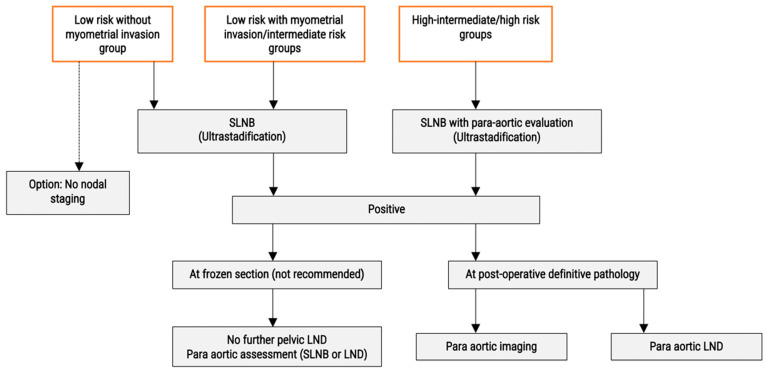
Proposed algorithms for the nodal management in early stages endometrial cancer. SNLB: Sentinel Lymph Node Biopsy; LND Lymph Node Dissection.

For the high-risk group, there is still a debate on whether SLN could replace the lymphadenectomy (LND). As LN involvement reaches about 20–25% in this situation [25], pelvic and para-aortic LN dissections have long been recommended. A complete lymphadenectomy procedure gives a 10–20% risk of lower-extremity lymphedema and a 10–25% risk of lymphocyst development [26,27,28,29,30,31]. However, if SLN minimizes radicality, it could also miss a positive LN, leading to under staging the EC. About 50% of patients with positive pelvic node have para-aortic LN involvement [32]. Finally, as for cervical cancer, several studies suggest that the presence of ITCs and micrometastases could be an important risk factor, with a need for adjuvant treatment, arguing for ultrastaging and the SLNB [11].

To move forward with SLNB in high-risk EC, some questions have to be answered. First, is there a therapeutic role of LND in high-risk EC? Or is it only for staging purposes? Additionally, second, what is the performance of the SLNB in this specific population?

In the retrospective study of Bendifallah et al., a significant impaired 5-year overall survival (OS) was found for patients with high-risk EC without lymphadenectomy, compared to patients with at least a pelvic lymphadenectomy. However, less adjuvant treatments were offered in the “no staged” group, which may explain the survival difference [33].

More recently, two retrospective studies have shown no survival difference between comprehensive LND and SLN dissection in high-risk groups, arguing for the absence of a therapeutic role, but instead a staging role of the LND [34,35]. These results have been confirmed in other histological subtypes. In the study of Bogani et al., comprising 113 non-endometrioid EC, similar disease-free and overall survival rates were found for patients receiving comprehensive lymphadenectomy and SLN alone.

In a retrospective study of Schiavone, no survival difference was found comparing SNLB to comprehensive LND in carcinosarcoma [36]. Similarly, in clear cell carcinoma, a trend towards impaired recurrence-free survival, without OS difference, was found in the SLNB group [37]. In these populations, most recurrences were distant or non-localized spreads, underlining more the need of systemic and radiation therapy than a comprehensive lymphadenectomy.

Concerning the performance of the SLNB in the high-risk group, two large prospective studies have confirmed the good performance of the SLNB in this setting, with a bilateral detection rate reaching 95%, and the para-aortic coverage of 80% [38,39].

In current ESGO/ESTRO/ESP guidelines, SLN dissection is an option for high-risk/high-grade EC. However, the issue of the diagnosis of isolated paraaortic nodes has been raised. After a cervical injection, most of the SLN are in the pelvic area. Some authors have described various techniques to improve the para-aortic detection rate in high-risk endometrial cancer: the “Two-step method” is a combination of a cervical and the bilateral uterine cornual areas with an 86% para-aortic detection rate [40,41]; the hysteroscopic peritumoral injection has shown an 88% para-aortic detection rate [42]; the transvaginal ultrasound-guided myometrial injection of radiotracer (TUMIR) has a lower 45% para-aortic detection rate [43,44].

An open question would ask which patients would be the best candidates for a surgical de-escalation, as molecular profiling provides novel insights into the molecular heterogeneity of EC. Four profiles are currently considered: DNA polymerase epsilon (POLE) mutations, mismatch repair proteins deficiency (MMRd), mutations in p53, and the copy number-low subtype (no surrogate marker).

The recent ESGO/ESTRO/ESP guidelines for the management of EC have integrated molecular biology into the definition of prognostic groups, and the adjuvant treatment decision. However, surgical strategies should also consider the integrated molecular risk assessment in the decision, especially in the LN-staging strategy.

Patients with a POLE mutation have a nodal involvement in less than 1% of cases, although apparent aggressive histopathologic features (about 65% of apparent pre-operative high-risk). Conversely, about 15% of TP53abn are associated with positive nodes [45,46].

As the surrogate markers of POLE, TP53abn and MMRd have already been validated to be reliable using pre-operative material [47], surgical strategies should consider the integrated molecular risk assessment in the decision to adapt the lymph node staging procedure. Lymphadenectomy should be considered in apparent low-risk EC if the molecular biology indicates a rather unfavourable prognosis. Likewise, the integrated risk assessment could help to identify “true” high-risk patients in which lymphadenectomy could contribute to longer survival, or, conversely, would have no benefit as chemotherapy is always indicated in these patients. Conversely, no staging could be proposed in “true” low-risk molecular cases (especially POLEmut).

Another concern is about the management of positive SLND. Frozen section is not recommended as it could impair the micrometastasis/ITC diagnosis, but in the case of pre-operative diagnosis of positive SLNB, pelvic LND should be abandoned, and the para-aortic area should be carefully examined, either by a SLND in this area, or by a para-aortic LND [48].

In case of a post-operative positive pelvic SLN, the open question is the reintervention for para-aortic LND. The balance is between missing a positive PA lymph node, and the morbidity of a reintervention in mostly overweight patients in a post-operative inflammatory period. Imaging has shown good performance in this setting, especially PET-CT, with sensitivity of 50–85%, specificity of 88–100% and VPN reaching 75–100% [49,50,51]. In the ESGO guidelines, in the case of positive lymph nodes, the staging could be either by surgery or imaging.

Finally, a systematic para-aortic irradiation in the case of pelvic positive lymph nodes could be proposed, whatever the imaging results. However, this could lead to an excess of morbidity and complicate the treatment in cases of recurrence, without improving the prognosis [52].

To conclude, SLNB is widely validated in endometrial cancer but it is necessary to clearly define the escalation and de-escalation field in the era of molecular biology. As this molecular classification seems efficient for predicting the benefit and indication of chemotherapy and pelvic irradiation, the main information of the nodal staging in the high-risk group is to indicate para-aortic irradiation. An effort to obtain an SLN in the para-aortic area has to be made in the high-risk EC. However, the real role of nodal gesture in EC is still questioned, as the cost effectiveness, potential harms, and prognosis benefit of SLNB compared to no-node dissection have never been established. The ongoing ENDO-3 study of Obermair et al. aims to answer this question [53].

## 4. Conclusions

The rise in the use of SLN in uterine cancers allowed us a de-escalation in the nodal gesture, but also in uterus management and the adjuvant treatment indication. This is just the beginning of the story. Several ongoing trials will give us important data in the coming years that could substantially change these propositions.

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
