# Peer review of "Sentinel Lymph Node Biopsy in Uterine Cancer: Time for a Modern Approach"

_cancers, 2023, doi:10.3390/cancers15020389_

Round 1
Reviewer 1 Report
This is a well written and important review on SLNB in uterine cancers. I appreciate the description of risks and benefits depending on stage and hitology. Also the diagrams are really helpful.
Authors should check:
Line 66-68: Sentence not complete
Line 74-75: Add references
Line 188-189: Check abbreviations. List at first mentioning
Author Response
Response to Reviewer 1 Comments
First, we would like to thank the reviewers for their comments.
In the revised version, we have modified the manuscript according to reviewers’ comment.
Point 1: Line 66-68: Sentence not complete
Response 1: We agree with the reviewer that this sentence is unclear. Actually, it was bullet points so the sentence was just an enumeration. But the bullet points have been removed in the current version, not by ourself…We added the bullet points and modifiy the sentence to “Postoperative radiation therapy after radical hysterectomy, is highly toxicic and should be avoid.”
Point 2 : Line 74-75: Add references
Response 2: The following references were added
- Kim, C.H.; Khoury-Collado, F.; Barber, E.L.; Soslow, R.A.; Makker, V.; Leitao, M.M.J.; Sonoda, Y.; Alektiar, K.M.; Barakat, R.R.; Abu-Rustum, N.R. Sentinel Lymph Node Mapping with Pathologic Ultrastaging: A Valuable Tool for Assessing Nodal Metastasis in Low-Grade Endometrial Cancer with Superficial Myoinvasion. Gynecol. Oncol. 2013, 131, 714–719, doi:10.1016/j.ygyno.2013.09.027.
- Ghoniem, K.; Larish, A.M.; Dinoi, G.; Zhou, X.C.; Alhilli, M.; Wallace, S.; Wohlmuth, C.; Baiocchi, G.; Tokgozoglu, N.; Raspagliesi, F.; et al. Oncologic Outcomes of Endometrial Cancer in Patients with Low-Volume Metastasis in the Sentinel Lymph Nodes: An International Multi-Institutional Study. Gynecol. Oncol. 2021, 162, 590–598, doi:10.1016/j.ygyno.2021.06.031.
- Bogani, G.; Mariani, A.; Paolini, B.; Ditto, A.; Raspagliesi, F. Low-Volume Disease in Endometrial Cancer: The Role of Micrometastasis and Isolated Tumor Cells. Gynecol. Oncol. 2019, 153, 670–675, doi:10.1016/j.ygyno.2019.02.027.
- Todo, Y.; Kato, H.; Okamoto, K.; Minobe, S.; Yamashiro, K.; Sakuragi, N. Isolated Tumor Cells and Micrometastases in Regional Lymph Nodes in Stage I to II Endometrial Cancer. J. Gynecol. Oncol. 2016, 27, e1, doi:10.3802/jgo.2016.27.e1.
Point 3 : Line 188-189: Check abbreviations. List at first mentioning
Response 3: The abreviation SLN was defined in line 33, and FS (Frozen section) was already defined on line 66.
Reviewer 2 Report
For cervical cancer:
1. I am confused because fertility sparing treatment can also be considered in other early stage cervical cancer (stage IA2, IB1, select IB2) rather the IA1 LVSI+ in the propose algorithm (Figure 1)
2. Do the authors consider assess the role of SLN based on the FIGO (2018) calssification (eg. IB1, IB2, IB3)
For endometrial cancer:
1. Selection criteria, sensitivity, specificity and predictive value had been well established for endometrial cancer. The proposed algorithm in Figure 2 added little to what is already known.
2. I agree with the authors that integration of molecular profiles (POLE, TP53, MMRd) into surgical strategies to be considered in staging for endometrial cancer.
3. I suggested the authors discussed the value and performance in high risk histology endometrial cancer, such as serous and clear cell carcinoma
Author Response
Point 1. I am confused because fertility sparing treatment can also be considered in other early stage cervical cancer (stage IA2, IB1, select IB2) rather the IA1 LVSI+ in the propose algorithm (Figure 1)
We totally agree with the reviewer. The text as well as the figure 1 were modified accordingly for stage IB1 without LVSI .
“In patients with no fertility wish, a one step procedure appears logical, with SLN biopsy, FS, and radical hysterectomy in case of negative FS. The place of systematic complementary pelvic dissection is unclear since diagnostic accuracy is good in this population. This strategy will make postoperative radiation therapy indicated in 10% of patients with false negative FS[9]. In patient patients with fertility wish, the alternative could be a two steps approach, with SLN biopsy and subsequent radical trachelectomy some days later, only in patients without macro or micrometastases. But this approach will expose all patients to two general anaesthesia, operative difficulties, and complications due to iterative parametrial dissections.”
However, for IB1 with LVSI and IB2, fertility sparing treatment ar not the standard of care, and were not included in our algorithm. We added a sentence to specify this point
“Fertility sparing treatment are not discussed for this stage, as it is not standard of care.”
Point 2. Do the authors consider assess the role of SLN based on the FIGO (2018) calssification (eg. IB1, IB2, IB3)
All the manuscript concerns the FIGO 2018. We specified that at the begining of the cervical cancer section. In this setting, the use of SLNB in the IB3 stage is not a standard of care.
Point 3. Selection criteria, sensitivity, specificity and predictive value had been well established for endometrial cancer. The proposed algorithm in Figure 2 added little to what is already known.
We agree with the reviewer that this figure does not add anything new. But we felt it was important to propose it, as a summary of the management. If the reviewer think that it is useless, we can remove it.
Point 4. I suggested the authors discussed the value and performance in high risk histology endometrial cancer, such as serous and clear cell carcinoma
We agree that the serous and clear cells histology were not enough discussed. We modified our manusript as follow :
“More recently, two retrospective study have shown no survival difference between comprehensive LND and SLN dissection in high-risk group, arguing for the absence of therapeutic but a staging role of the LND [34,35]. These results have been confirmed in other histological subtypes. In the study of Bogani et al., comprising 113 non-endometrioid EC, similar disease-free and overall survival were found for patients receiving comprehensive lymphadenectomy and SLN alone.
In a retrospective study of Schiavone, no survival difference was found comparing SNLB to comprehensive LND in carcinosarcoma[36]. Similarly, in clear cell carcinoma, a trend to an impaired recurrence free survival, but without OS difference was found in the SLNB group[37]. In these population, most recurrences were distant or non-localized spread, underlining more the need of systemic and radiation therapy, than a comprehensive lymphadenectomy.”
Reviewer 3 Report
The article is very well analyzed and meets the requirements of the journal for publication, so I recommend it for publication after minor revision:
1. The font of Figure 1 is obviously larger than that of Figure 2 and Figure 3, and it is recommended to keep the same font size.
2. The authors reviewed the standards and pending questions of SLNB in cervical and endometrial cancer, according to published data and on-going trials. But I think it is not enough, there are some prospects and suggestions to solve the question of SLNB.
Author Response
Point 1. The font of Figure 1 is obviously larger than that of Figure 2 and Figure 3, and it is recommended to keep the same font size.
We thank the reviewer for this point. We modified the figures to be with same font and size
Point 2. The authors reviewed the standards and pending questions of SLNB in cervical and endometrial cancer, according to published data and on-going trials. But I think it is not enough, there are some prospects and suggestions to solve the question of SLNB.
As suggested, we have added a paragraph to discuss the future of the SLNB, especially the abstention of nodal gesture in stage I.
”But the real role of nodal gesture in EC is still questioned, as the cost effectiveness, potential harms and prognosis benefit of SLNB compared to no-node dissection has never been established. The ongoing ENDO-3 study of Obermair et al aims to answer this question"